# Evaluation of a New DNA Extraction Method on Challenging Bone Samples Recovered from a WWII Mass Grave

**DOI:** 10.3390/genes15060672

**Published:** 2024-05-23

**Authors:** Barbara Di Stefano, Irena Zupanič Pajnič, Monica Concato, Barbara Bertoglio, Maria Grazia Calvano, Solange Sorçaburu Ciglieri, Alessandro Bosetti, Pierangela Grignani, Yasmine Addoum, Raffaella Vetrini, Francesco Introna, Serena Bonin, Carlo Previderè, Paolo Fattorini

**Affiliations:** 1Department of Medicine, Surgery and Health, University of Trieste, 34127 Trieste, Italy; barbarbarbara.distefano@phd.units.it (B.D.S.); monica.concato@phd.units.it (M.C.); soleburu@hotmail.it (S.S.C.); yasmine.addoum@studenti.units.it (Y.A.); raffaella.vetrini@studenti.units.it (R.V.); sbonin@units.it (S.B.); fattorin@units.it (P.F.); 2Institute of Forensic Medicine, Faculty of Medicine, University of Ljubljana, 1000 Ljubljana, Slovenia; irena.zupanic@fm.uni-lj.si; 3Section of Legal Medicine and Forensic Sciences, Department of Public Health, Experimental and Forensic Medicine, University of Pavia, 27100 Pavia, Italy; barbara.bertoglio01@universitadipavia.it (B.B.); pierangela.grignani@unipv.it (P.G.); 4Section of Legal Medicine, Interdisciplinary Department of Medicine (DIM), University-Hospital of Bari, Giulio Cesare Square 11, 70124 Bari, Italy; mg.calvano@gmail.com (M.G.C.); francesco.introna@uniba.it (F.I.); 5Promega Italia, 20126 Milano, Italy; alessandro.bosetti@promega.com

**Keywords:** bones, DNA degradation, DNA typing, mass grave, disaster victim identification

## Abstract

Bones and teeth represent a common finding in ancient DNA studies and in forensic casework, even after a long burial. Genetic typing is the gold standard for the personal identification of skeletal remains, but there are two main factors involved in the successful DNA typing of such samples: (1) the set-up of an efficient DNA extraction method; (2) the identification of the most suitable skeletal element for the downstream genetic analyses. In this paper, a protocol based on the processing of 0.5 g of bone powder decalcified using Na_2_EDTA proved to be suitable for a semi-automated DNA extraction workflow using the Maxwell^®^ FSC DNA IQ™ Casework Kit (Promega, Madison, WI, USA). The performance of this method in terms of DNA recovery and quality was compared with a full demineralisation extraction protocol based on Qiagen technology and kits. No statistically significant differences were scored according to the DNA recovery and DNA degradation index (*p*-values ≥ 0.176; r ≥ 0.907). This new DNA extraction protocol was applied to 88 bone samples (41 femurs, 19 petrous bones, 12 metacarpals and 16 molars) allegedly belonging to 27 World War II Italian soldiers found in a mass grave on the isle of Cres (Croatia). The results of the qPCR performed by the Quantifiler Human DNA Quantification kit showed values above the lowest Limit of Quantification (lLOQ; 23 pg/µL) for all petrous bones, whereas other bone types showed, in most cases, lower amounts of DNA. Replicate STR-CE analyses showed successful typing (that is, >12 markers) in all tests on the petrous bones, followed by the metacarpals (83.3%), femurs (52.2%) and teeth (20.0%). Full profiles (22/22 autosomal markers) were achieved mainly in the petrous bones (84.2%), followed by the metacarpals (41.7%). Stochastic amplification artefacts such as drop-outs or drop-ins occurred with a frequency of 1.9% in the petrous bones, whereas they were higher when the DNA recovered from other bone elements was amplified (up to 13.9% in the femurs). Overall, the results of this study confirm that petrous bone outperforms other bone elements in terms of the quantity and quality of the recovered DNA; for this reason, if available, it should always be preferred for genetic testing. In addition, our results highlight the need for accurate planning of the DVI operation, which should be carried out by a multi-disciplinary team, and the tricky issue of identifying other suitable skeletal elements for genetic testing. Overall, the results presented in this paper support the need to adopt preanalytical strategies positively related to the successful genetic testing of aged skeletal remains in order to reduce costs and the time of analysis.

## 1. Introduction

Bones and teeth are the most resistant components in the human body, and they represent the usual finding even after a long burial. Genetic typing is the gold standard for personal identification of skeletal remains [1,2]. Furthermore, considering that genetic testing of skeletal elements is expensive and time consuming, it is strongly recommended to select suitable bone elements [3]. In fact, assuming the absence of PCR inhibitors such as humic acids [3] in DNA extracts, the success of DNA typing depends mainly on the amount and the quality of the genetic material available [3,4,5,6,7,8,9,10,11].

Bone taphonomy is a complex process in which both the type of skeletal elements chosen and environmental factors play key roles [3,12]. Although no strict correlation between bone tissue preservation and DNA preservation has been found so far, it is commonly known that skeletal elements from different anatomical regions or even different portions of the same bone element may produce different results [3,4,5,6,7,8,9,10,11,12,13,14,15,16,17,18,19,20,21]. The reasons underlying different outcomes have been evaluated in a few studies and indicate that bone density could be a protective factor in DNA integrity [22]. It is also well established that the bone element offering the highest DNA yields and quality is the temporal bone, particularly the inner ear part of the petrous bone [15,16,17,18,19,20,21,23,24]. When the petrous bone is not available, other skeletal elements, such as long bones (mainly femurs and tibias) and intact teeth, are to be preferred for genetic testing [3,19]. More recently, metacarpal bones [21] and vertebrae [13] were described to provide good chances of typing results in forensics. In order to confirm the usefulness of short bone elements, these findings should be evaluated in larger sets of bones buried in different environmental conditions. A standard forensic approach is to collect as many bones as possible from the same skeleton [4,5,6,7,8,9,10,11,14]. Eventually, case-to-case strategies need to be adopted in relation to the number of skeletons found, the post mortem interval, the feasibility of transporting all skeletal remains, the facilities available in the laboratory for storing the remains, etc.

Another crucial factor for successful typing is the DNA extraction protocol. As recently reviewed [14], several DNA extraction protocols were developed in the last decade with silica-based or magnetic silica-based bead procedures replacing the traditional phenol/chloroform organic purification. Moreover, even if several different approaches have been proposed [14,25], it is generally accepted in forensics that Na_2_EDTA-decalcified bone powder provides higher yields of DNA than the un-decalcified one [3,14]. Recently, commercial kits specifically designed for DNA extraction from 50 to 150 mg of bone powder have been available to scientists and used in forensic casework analyses [14,26,27,28,29,30]. However, when challenging bone samples are processed, larger amounts of bone powder may be required for a successful PCR-CE-based typing. In these cases, it may also be necessary to plan alternative strategies according to the laboratory’s instrumentations and the number of samples to be processed [3,14].

Recently, we were involved in a forensic investigation of the skeletal remains found in a Second World War (WWII) mass grave which, according to historical records, belonged to 27 male Italian soldiers. To process the request promptly, it was decided to move from the manual preparation of the samples [31] to the new semi-automated workflow described below. In addition, as this method was applied to a set (*n* = 88) of samples, the preliminary results of the DNA typing are illustrated.

## 2. Materials and Methods

This paper describes the modifications introduced to the standard protocol by increasing to 0.5 g the amount of bone powder processed using the Maxwell^®^ FSC DNA IQ™ Casework Kit (Promega, Madison, WI, USA). The experiments were performed in one laboratory (Lab A), and the resulting DNA outcomes were compared to those made with a highly performing DNA extraction method [32] set up in another laboratory (Lab B) also involved in the present study. The protocol of Lab B was based on the DNA extraction from the same amount of bone powder (500 mg) using Qiagen technology and kits [13,21,33,34].

Details on the optimization of the proposed protocol are reported in Appendix A.

### 2.1. Bone Samples

The skeletons were exhumed in May 2019 from a mass grave on the island of Cres (Croatia). According to historical records, the skeletal remains belonged to 27 Italian soldiers. No clothes or uniforms were found, confirming the information on record which reported the soldiers to have been executed naked on April 21, 1945. The mass grave was located at sea level, and the remains were buried about 1–2 m under the ground surface. The island of Cres is located in the north-east Adriatic Sea (44°42′ N; 14°24′ E), and the calcareous soil facilitates the flow of rainwater. The climate is mild, with average temperatures ranging from 7.9 °C in winter to 23.1 °C in summer; the rainfall is 838 mm/year.

Twenty-seven metal boxes were handed over to the Italian government on 13 November 2019. The anthropological and medico-legal examination performed in 2022 by the University of Bari (Italy) showed that the skeletal remains belonged to more than 27 subjects because 29 right human femurs were scored. The skulls were always fractured with disarticulation of the temporal bones; in addition, in a few boxes, the bones appeared commingled, i.e., belonging to more than one subject (paper in preparation). For the genetic testing, bones and teeth were sampled for a total of 341 specimens. Strict precautions to prevent contamination [32,35,36,37] were adopted from this step only. The specimens were then delivered to the University of Trieste (Italy), where they were stored at room temperature, in the dark, until molecular analysis.

For the present study, all the right femurs (*n* = 29) and all the right petrous bones (n = 19) were selected, as well as 12 left femurs, 12 metacarpals and 16 molar teeth (with no sign of caries), for a total of 88 samples (see Table A4 of Appendix B). The criterion adopted was to analyse at least two samples for each skeleton. Following the indication of previous studies, we used the bone type tissues that offered the best chances of genetic typing, i.e., the compact portion of the diaphysis for the femurs [33], the compact portion of the epiphysis for metacarpals [21] and the inner ear part for petrous bones [20]. Lastly, given that there is no DNA in enamel, the latter was removed from the 12 molars selected, and the remaining entire teeth were then used for DNA extraction [14].

### 2.2. Bone Cleaning and Pulverisation

All the procedures described in this paragraph were performed in a room designed exclusively for processing old skeletal remains. To remove soil, the surfaces of the bones and teeth were cleaned mechanically using brushes and rotary sanding tools (for petrous bones, further mechanical cleaning was conducted after isolation of the inner part). To remove exogenous DNA contamination [3,14], the tooth or approximately 1.5 g of each bone sample was incubated in 0.5% sodium hypochlorite for 4 min with gentle agitation, washed in sterile bi-distilled water three times and dried at room temperature overnight. In addition, the surfaces of the samples were UV-radiated for 5 min before pulverisation.

Bones and teeth were pulverised at 30 Hz for 1–2 min [32] using a MM 400 Planetary Ball Mills (Retsch, Haan, Germany) equipped with metal grinding vials of 25 mL and metal balls with Ø of 16 mm (Verder, Castleford, UK). Liquid nitrogen was used to prevent the heating of the samples [32]. The 88 powder samples were stored in 50 mL Falcon tubes at room temperature in the dark until molecular analysis.

Bone fragments from the 12 left femurs were collected and sent to Lab B for bone pulverisation and DNA extraction, according to the workflow shown in Figure 1. For pulverisation, Lab B used a Bead Beater MillMix 20 (Tehtica-Domel, Železniki, Slovenia) equipped with metal grinding vials of 25 mL and metal balls with Ø of 20 mm (Tehtica-Domel, Livonia, MI, USA); 30 Hz for 1–2 min was applied [32]. After pulverisation, the samples were processed immediately.

### 2.3. Bone Decalcification and DNA Extraction

The protocol described below was applied to all 88 powder samples. In detail, 0.5 g of each powder sample was decalcified by mixing it with 15 mL of Na_2_EDTA pH 8.0 in 50 mL Falcon tubes at 40 °C with shaking at 850 rpm overnight. After centrifugation (2800 rpm for 15 min) and two washes with 10 mL of water, the pellet was resuspended by adding 460 µL of extraction buffer (1.2% SDS, 10 mM Tris pH 8.0, 10 mM Na_2_EDTA pH 8.0 and 100 mM NaCl), 40 µL of 1 M DTT and 30 µL Proteinase K (20 mg/mL). After incubation overnight at 37 °C at 850 rpm, the samples were added with 40 µL Proteinase K (20 mg/mL) and incubated at 56 °C for 4 h at 850 rpm. At the end of this second incubation, the samples were transferred into 1.5 mL Eppendorf tubes and centrifuged at 14,000 rpm for 3 min. An amount of 800 µL of the sample was mixed with 400 µL of the Lysis Buffer (LB) provided with the Maxwell^®^ FSC DNA IQ™ Casework Kit (Promega) and vortexed for ten seconds. The whole sample (1.2 mL, twice the volume recommended in the protocol) was loaded into the cartridge and processed in a Maxwell RCS TM441 (Promega) apparatus following the recommended procedures. The DNA IQ Casework protocol, freely available at www.promega.com, was followed. DNA was eluted in a final volume of 60 microliters. Negative extraction controls (NECs) were processed every 6–12 bone samples. To assess the procedure repeatability, the bone powder from six random petrous bones was used for DNA extraction by a different operator three months after storage at room temperature in the dark.

DNA extractions were also carried out in Lab B using the same amount of bone powder (0.5 g) and the Qiagen technology and protocols described by Zupanič Pajnič [32]. At the end, 18 bone powders prepared in Lab A and 12 bone powders prepared in Lab B were used to compare the DNA extraction efficiencies, as shown in Figure 1.

### 2.4. DNA Quantification

The DNA samples were quantified in duplicate by qPCR. Lab A used the Quantifiler Human DNA Quantification kit (ThermoFischer Scientifics; TFS, Waltham, MA, USA) following the conditions detailed in ref. [31]. This kit allows the simultaneous detection of a 62 bp long single-copy target within the telomerase reverse transcriptase gene (hTERT) and of an IPC (Internal Positive Control). The LOQ (Limit of Quantification) ranged from 0.023 ng/µL to 50 ng/µL; the LOD (Limit of Detection) was set previously at 0.001 ng/µL [31]. Lab B used the PowerQuant kit (Promega) following the conditions detailed in ref. [33]. This kit allows the simultaneous amplification of four targets: three multi-copied targets, that is, the Auto (84 bp long), the Y-specific (134 bp long) and the Deg (294 bp long), and the IPC. The LOQ ranged from 0.0032 ng/µL to 50 ng/µL both for the Auto and the Y target, whereas the lowest LOQ for the Deg was 0.0005 ng/µL; the LOD was set previously at 0.0001 ng/µL for all quantification targets [33].

### 2.5. DNA Typing

Only samples with detectable amounts of DNA in at least one of the two assays with Quantifiler were used for genotyping. This resulted in the genotyping of 52 out of 88 samples by the employment of the PowerPlex Fusion (Promega) kit, which simultaneously amplifies 22 autosomal STR markers and 2 Y-specific targets (Amelogenin and DYS391). In detail, all 19 samples from the petrous bones, 11 samples from the right femurs, 6 from the left femurs, 11 from the metacarpals and 5 samples from the teeth were amplified in the recommended final volume of 25 µL. A total of 30 PCR cycles were run for the samples with 0.5–1.0 ng of template, whereas 32 cycles were run for the samples with less than 0.5 ng of DNA (in those cases, 15 µL of DNA extract, i.e., the maximum volume allowed was loaded into the PCR tube). Negative and positive PCR controls as well NECs were analysed. The samples were then run through a 310 ABI automatic DNA sequencer (Applied Biosystems, Waltham, MA, USA), and the resulting data were analysed with the GeneMapperID^®^ ver 3.2.1 software (Applied Biosystem). Analytical and stochastic thresholds were set at 50 and 150 rfu, respectively. After the first round of amplification, only samples with at least twelve autosomal STR markers successfully amplified underwent duplicate tests (this procedure was not performed for profiles matching other profiles already loaded in the database). Thereafter, only samples for which it was possible to generate “*consensus*” profiles [38] for at least 12 autosomal STR markers were considered “*suitable for personal identification*” [8] and classified as full (22/22 autosomal markers) and partial profiles (≥12 autosomal markers), respectively. Stochastic amplification artefacts such as allele drop-out and/or allele drop-in were scored by cross-checking the results of the replicated amplifications.

### 2.6. Data Analysis

Statistical analyses were carried out with the Stata/SE 16.0 package (StataCorp, College Station, TX, USA) and a *t*-test or ANOVA (when indicated) was used for calculation. Significance was assumed for a *p*-value < 0.05.

### 2.7. Exclusion Database

All the individuals involved in any step of the molecular analysis were genotyped following standard procedure. Informed consent was acquired before saliva sampling.

## 3. Results

### 3.1. Evaluation of the Extraction Method

The new protocol includes sample extraction through two incubation steps (at 37 °C and 56 °C) followed by automated DNA purification using the Maxwell RCS TM441 apparatus. As reported in Appendix A, a volume double the one suggested in the kit user’s manual (www.promega.com) (accessed on 8 May 2022) (1.2 mL, which is 0.8 mL of sample added to 0.4 mL of Lysis Buffer provided in the kit) was loaded into the Maxwell^®^ FSC DNA IQ™ Casework Kit cartridge. Even if this modification implies a lower % recovery (about 20–25%) than when following the recommended procedure, the total recovery of DNA is higher (up to 50–55%).

The recovery (ng DNA/g bone) of this DNA extraction method was compared with the protocol routinely employed in Lab B [32] by processing the same 18 bone powders. Figure 2 compares the recovery of the two methods as assessed by the 84 bp long Auto probe of the PowerQuant kit. No statistically significant difference was found (*p*-value = 0.709; r = 0.912). It has to be noted, however, that high amounts of DNA were not efficiently recovered in Lab A, likely due to the saturation of the capacity of the magnetic beads to bind the DNA (this outcome was observed even during the optimisation of the protocol; see Table A3).

The 294 bp long Deg probe showed no Cq values in two and five femur samples extracted in Lab A in Lab B, respectively. As shown in Figure 2, however, the quantification with the Deg probe did not show any difference (*p*-value = 0.176 r = 0.907) between the two laboratories. The Auto/Deg ratio, an indicator of the degradation extent, ranged from 5.5 to 418 and from 6.4 to 311 in Lab A and Lab B, respectively, but no statistically significant differences were observed between the samples extracted in the two laboratories (*p*-value = 0.548; r = 0.958). Finally, no differences were found for the IPC values (*p*-value = 0.754), showing that both protocols efficiently removed the PCR inhibitors.

Figure 3 shows the normalised yields of DNA recovered from the two skeletal elements chosen at this step of the study, the femur and the petrous bone. The petrous bones yielded—on average—about 182-fold higher amounts of DNA than the femurs (median values: 52.9 ng vs. 0.143 ng and 59.1 ng vs. 0.150 ng in Lab A and Lab B, respectively). Higher recoveries from petrous bones are perfectly in agreement with previous data showing that those skeletal elements represent the best source of DNA from human bones [3,14], likely due to their intrinsic anatomical [23] and physiological features [22].

Since the pulverisation procedure, mainly the fineness of the bone powder [27] and the heat developed during the grinding procedure [35,39], can also affect DNA recovery, 12 femur fragments were pulverised and extracted in Lab B, as shown in Figure 1. The comparison of the DNA recovery among samples processed in different laboratories showed a *p*-value of 0.666, as assessed by the ANOVA test. This result rules out pulverisation in Lab A as a procedure capable of interfering significantly with efficient DNA recoveries, which leads to the conclusion that the femurs found in the mass grave actually contained very low amounts of DNA (on average, no more than 0.3 ng/g of bone powder). As reported in previous work, up to 3.2 nanograms of DNA/gram of bone powder was recovered, on average, from a set of 69 femurs found in a different WWII mass grave [33].

Lastly, two operators extracted the same six powders from petrous bones in Lab A, at different times. Although the number of samples is small, and the Quantifiler kit provided data only on quantifying a 62 bp long target, good repeatability (*p*-value = 0.291; r = 0.902) was found over time (three months of storage at room temperature in the dark). The results of the recoveries (nanograms of DNA/gram of bone powder) are reported in Figure A1.

PCR inhibition was never detected when the Cqs of the IPC probes of the two kits were evaluated.

### 3.2. DNA Quantification of the Bone Samples

The Quantifiler (TFS) kit was used to assess the DNA amount recovered from the 88 bone samples extracted with the method described in the present study (Lab A). The results are summarised in Table 1.

The petrous bone always provided quantification values higher than the lowest LOQ (23 pg/µL), with a mean amount of 440 pg/µL. Out of the remaining bone types, only the metacarpal and the femur showed values within the range of the LOQ (in 16.6% and 3.4% of the tests, respectively). Detectable levels of DNA (that is, LOD ≥ 1 pg/µL) were, however, found in all bone types, with a frequency ranging from 15.6% (tooth) to 58.3% (metacarpals). No inhibition was detected, as shown by the Cq of the IPC [40,41].

### 3.3. Genetic Typing of the Challenging Bone Samples

To this aim, only bone elements with detectable levels of DNA in at least one of the two qPCR assays were selected. The results yielded from 78 PCR tests (excluding the DNA control samples) are reported in Table 1. Genetic typing from the petrous bone samples was always achieved, with full profiles (22 out of 22 autosomal STR markers) in 84.2% of the tests (see Figure 4). Six PCR tests gave partial profiles characterised by 16 markers/electropherograms (median value). As shown in Figure A2, a typical ski-slope profile was obtained from these challenging samples, in agreement with the scored degradation levels [35].

The other bone elements were successfully typed with different rates (from 20% to 83.3% for teeth and metacarpals, respectively). Only five, one and one full electropherogram was achieved for the metacarpal, tooth and femur, respectively. Partial profiles characterised by 18–20 markers/electropherograms (median values) were obtained in 16 tests, whereas the remaining 17 tests showed 5 markers/electropherograms (median value; min = 0; max = 9) and were then classified, such as “*no typing*” (see Figure 4). Remarkably, the petrous bone provided the same genotypes in the two PCR replicates for 98.1% of the markers, whereas the other bone types yielded profiles affected by stochastic artefacts in up to 13.9% of the replicates likely originated by high degradation levels and/or low amounts of template used for the PCR amplification [35].

As discussed above, *consensus* profiles [38] for at least twelve autosomal STR markers [8] in duplicated experiments were considered “*suitable for personal identification*” [8]. The *consensus* [38] approach allowed the identification of 22 different STR profiles suitable for comparisons (see Table A4 of Appendix B). Out of them, 17 were full profiles, whereas the remaining 5 were partial, with at least 16 scored markers. Importantly, bones found in the same metal box did not match in three cases. However, all results of the genetic typing can be considered useful for comparison with reference samples (relatives of the alleged soldiers) we are still trying to identify and collect.

No amplicons were scored in the PCR and extraction negative controls. All the profiles were compared with the ones stored in the laboratory exclusion database, and no matches were identified.

## 4. Discussion

When the skeletal remains collected in the mass grave in the isle of Cres were delivered to our lab, there were two main goals to achieve: the first was to set up a reliable protocol for DNA extraction in order to maximise the DNA extraction efficiency from these very challenging skeletal remains, and the second was to select the best-performing bone elements for individual genetic identification.

To this aim, we first used the Bone DNA extraction kit (Promega) on 150 mg of bone powder from femurs. As this method allowed no or very poor qPCR quantification values, we decided to increase the amount of starting bone powder to 0.5 g and set up a protocol based on the employment of the Maxwell^®^ FSC DNA IQ™ Casework Kit (Promega). The main advantage of this semi-automated DNA extraction protocol compared to other protocols, some of which are based on organic purification [3,14,31,42,43], is the decrease in the risk of human error and cross-contamination of the samples due to the automation of the DNA purification steps, especially by reducing physical manipulations of the specimens. The bone lysates, in fact, are transferred from 50 mL Falcon tubes into the cartridge after three simple and quick steps: (1) short centrifugation (in an Eppendorf tube) to pellet the undigested bone powder (if any); (2) transferring the sample to a second Eppendorf tube and mixing it with the buffer provided in the kit; (3) loading the sample into the prefilled cartridge.

The performances of the above-described protocol in terms of DNA recovery and the level of DNA degradation were compared with those of an optimised DNA extraction protocol from bone samples based on the Qiagen technology and kits [32]. The comparison did not show any statistically significant difference between the two protocols; for this reason, the one described in this paper can be efficiently performed in any laboratory equipped with a Maxwell automated DNA extraction apparatus. Although it is not a real issue, one limit to the present method is the low repeatability when the yields of DNA are higher than 30–50 nanograms, likely due to the saturation of the magnetic beads.

The DNA extraction protocol was applied to 88 specimens from different bone elements (72) and 16 teeth (molars) selected for this study. The results of this preliminary report confirm that petrous bone is the bone element providing the best chances for successful DNA typing. The high amounts of DNA recovered from this bone element is the simplest but most convincing explanation for the successful typing scored in all 19 petrous bone samples considered in this study [41].

The other bone samples seemed to be more compromised, as assessed by a lower amount of DNA recovered and a lower quality of the resulting genetic profiles. In fact, according to the sensitivity of the qPCR assay used, detectable amounts of DNA were found in approximately 25% of the qPCR tests performed on femur samples, in 58% of tests performed on metatarsal samples and in 16% of the tests performed on tooth samples (see Table 1). In forensic genetics, qPCR data are used mainly to address the operator on the strategies to be adopted in downstream typing approaches (volume of sample to be added to the PCR reaction and number of PCR cycles) [1,35]. It is well known that the use of different quantification kits can result in different quantification values for many reasons (length and molecular sequence of the targets, chemistry, calibration standards, etc.) [40]. In addition, each kit has its own sensitivity (for example, the LOD of the PowerQuant is about 10-fold lower than that of the Quantifiler kit). Thus, even if other commercial kits could likely provide quantification values for part of the bone samples analysed in this study which showed no Cq in the Quantifiler assay, it is also true that sub-cellular amounts of DNA are of scarce utility for a reliable STR-CE typing. However, as recently pointed out for historical bone samples aged from 80 to 800 years [41], it is clear that the number of scored markers is positively linked to the amount of the template used for the genetic testing, both for traditional PCR-CE and next-generation sequencing approaches.

In agreement with the limited amounts of template available for STR amplification, the genetic typing showed low percentages of successful typing. Overall, our results partially agree with previous data highlighting that metatarsals are ideal bone elements for genetic typing [21], even if a possible explanation for our finding could be ascribed to the peculiar burial conditions of the mass grave of the isle of Cres [3,14,44,45,46,47]. In addition, the low-quality genetic results obtained for molar teeth in our study do not support the positive outcome described for these elements recovered from some mass graves [14,38,42]. Again, as reported in several studies [3,12,14,44,45,46,47], environmental burial conditions (i.e., temperature and soil properties such as pH and ionic composition, hydrology, etc.) are key factors for biomolecule preservation.

The availability of genetic profiles based on at least 12 STR markers is recommended for successful individual identification in DVI (Disaster Victim Identification) casework [8]. However, it is an obvious consideration that the availability of more genotyping data can increase the chances of successful identification, in particular if the reference samples (ante mortem) are not first-degree relatives. Figure A1 provides an overview of the results achieved up to now. Seventeen full profiles (based on 22 autosomal STR markers) and five partial profiles with at least 16 markers were uploaded into our post mortem databases. Thus, the results presented in this paper seem to be promising and suggest that the analysis of other bone specimens collected from other skeletal elements could allow the recovery of all genetic profiles belonging to the 27 Italian soldiers whose bodies are likely to have been buried in that mass grave.

The anthropological and medico-legal examination of the 27 metal boxes handed over to the Italian government showed that the skeletal remains belonged to at least 29 subjects, and the bones were also likely commingled; still, only 19 right petrous bones were found within the metal boxes. DNA typing provided definitive evidence of commingled remains in three cases for which a comparison was performable. Therefore, our results highlight the need for accurate planning of the DVI operation, which should be carried out by a multi-disciplinary team. In particular, the added value of a forensic anthropologist should allow for collecting all the bone elements/fragments and for dealing with commingled scenarios [45], therefore reducing the effort for genetic typing.

A final consideration is that, while it is well known that an accurate selection of bone elements for DNA analysis is fundamental for successful genetic typing [3,14], the selection needs to be customised for each specific DVI scenario. In other words, while it is true that petrous bone usually outperforms other bone elements [15,16,17,18,19,20,21,46,48], the identification of other suitable skeletal elements seems to be a case-to-case tricky matter, depending on the features of the mass grave and the environmental conditions where the skeletons were buried [3,12,14,41,44,45,46,47]. This implies that an expensive and time-consuming strategy based on multiple samplings of different bone elements remains the only way to obtain genetic profiles from challenging skeletal remains [4,11,12,13,14,15,16,17,18,19,20,21], especially in DVI casework, as recommended by the DNA commission of the International Society for Forensic Genetics [8]. Thus, even the results presented in this paper highlight the need to search preanalytical parameters (such as bone density, bone composition and micro-computed tomography) positively related to DNA profiling, as performed in a few pilot studies performed on bones of archaeological [22,46,49,50] and forensic interest [51]. In conclusion, a multi-disciplinary approach is irreplaceable in the genetic analysis of DVI.

## 5. Conclusions

In this work, we described the set-up of a new protocol based on the processing of 0.5 g of decalcified bone powder whose DNA recovery was similar to a well-established protocol [32]; therefore, it was applied to a selected sample of bones found in a WWII mass grave. Although the total number of samples is limited (*n* = 88) and the qPCR quantification was conducted with a kit having a (relatively) low sensitivity, the results confirm that petrous bone should be preferred for genetic typing. Our results showed that petrous bone—likely due to its intrinsic properties—outperforms other bone elements (femurs, metacarpals and teeth) in quantity and quality. Overall, the results of this study support that burial conditions play a fundamental role in biomolecule preservation, and no generalisation is allowed. Femurs, which usually provide satisfying results [3,14,38,42], yielded approximately ten-fold less DNA than other femurs found in a different WWII mass grave studied by us [33]. Similarly, teeth found in the mass grave of the isle of Cres yielded low levels of DNA. Teeth are the hardest organs in the human body, with the enamel protecting the crown; still, in spite of their anatomical features and the absence of dental caries, they did not escape degradation processes. More encouraging, instead, were the results from metacarpals. However, further studies are needed to understand the role of environmental factors in DNA preservation within different types of bone remains.

## Figures and Tables

**Figure 1 genes-15-00672-f001:**
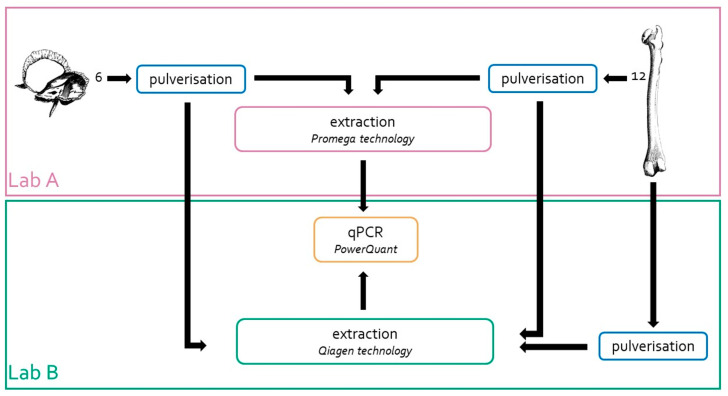
Strategy adopted to assess the efficiency of the extraction method developed in Lab A. The same powders from eighteen bone samples (6 from petrous bone and 12 from femur) were extracted in two different laboratories (Lab A and Lab. B). In addition, 12 femur samples were pulverised and extracted in Lab B. DNA quantification of all these samples was carried out only in Lab B by using the PowerQuant kit.

**Figure 2 genes-15-00672-f002:**
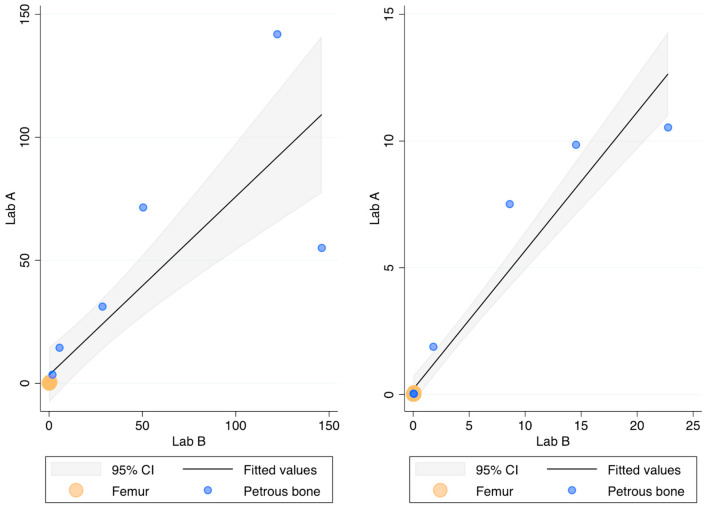
DNA recovery as assessed by the Auto probe (**left**) and the Deg probe (**right**) of the PowerQuant kit; *x*-axis: ng DNA/g bone in Lab B; *y*-axis: ng DNA/g bone in Lab A.

**Figure 3 genes-15-00672-f003:**
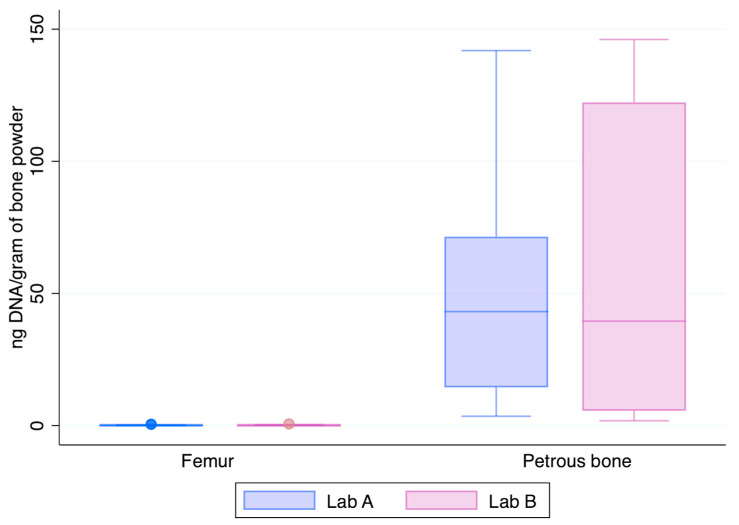
DNA recovery as assessed by the Auto probe of the PowerQuant kit from the 12 femurs and 6 petrous bones (x-axis); y-axis: nanograms of DNA/gram of bone powder.

**Figure 4 genes-15-00672-f004:**
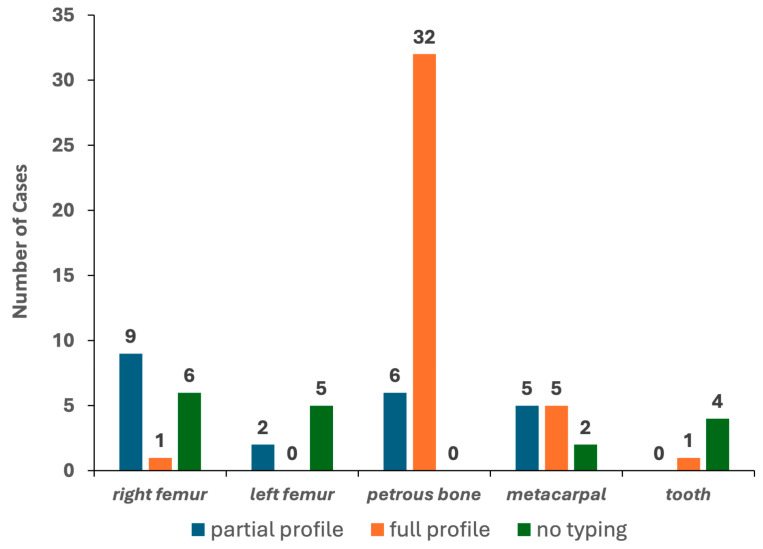
Results of the 78 PCR tests performed with the PowerPlex Fusion kit.

**Table 1 genes-15-00672-t001:** qPCR and STR typing results from different bone types. n: number of DNA samples; >LOD: percentage of qPCR tests above the Limit of Detection; lLOQ: percentage of qPCR tests above the lowest Limit of Quantification; pg/µL: average (±standard deviation) of DNA concentration as assessed by the Quantifiler kit; in brackets, the median (in bold), the minimum and the maximum values; STR typing: number of PCR tests with successful STR typing (that is, ≥12 markers) out of the total number of PCR tests (the percentage is in the bracket); markers: median number of markers scored in the partial profiles; S.A.: percentage of stochastic PCR artefacts (drop-outs and/or drop-ins) in the replicates; n.a.: not applicable.

Bone Type	n	>LOD	>lLOQ	pg/µL	STR Typing	Markers	S.A.
Right femur	29	21.7%	3.4%	5.9 ± 16.8 (***0***, 0, 88)	10/16 (62.5%)	20	13.9%
Left femur	12	25.0%	0%	1.2 ± 2.3 (***0***, 0, 7)	2/7 (28.5%)	18	8.3%
Petrous bone	19	100%	100%	440 ± 343 (***401***, 118, 1.179)	38/38 (100%)	16	1.9%
Metacarpal	12	58.3%	16.6%	5.0 ± 7.2 (***0***, 0, 27)	10/12 (83.3%)	18	2.3%
Tooth	16	15.6%	0%	1.0 ± 2.2 (***0***, 0, 9)	1/5 (20.0%)	-	n.a.

## Data Availability

Data are contained within the article. Other data presented in this study are available on request from the corresponding author.

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
