# Peer review of "Evaluation of a New DNA Extraction Method on Challenging Bone Samples Recovered from a WWII Mass Grave"

_genes, 2024, doi:10.3390/genes15060672_

Round 1

Reviewer 1 Report

Comments and Suggestions for Authors

I would like to extend my congratulations to the authors of the article for their objective analysis and clear presentation of the subject matter. I will now proceed to provide some notes on the work.

1. Did the authors pass an ethical committee for the human bone analysis, since this involves sample destruction from war victims?

2. Before bone pulverization, one of the fundamental steps is the cleaning of the samples. Petrous bones accumulate a lot of soil in their interior. How did the authors clean the samples since this can affect all the analysis, dragging inhibitors from the soil...

3. line 212: "ANOVA was used for calculation". Authors must specify what calculation was performed.

4. Figure 2 is not intelligible due to its poor quality.  

5. Line 389-393: environmental conditions are indeed key to DNA preservation. Authors use this argument to justify the absence of results for teeth samples. However, petrous bones were in the same conditions. How do authors explain that the genetic material from teeth, which are theoretically protected by enamel, did not produce results, and bone samples did?

Reviewer 2 Report

Comments and Suggestions for Authors

Dear Authors,

The manuscript “Ealuation of a new DNA extraction method on challenging  bone samples recovered from a WWII mass grave ” by Barbara Di Stefano et al. provides an interesting dose of analytical knowledge regarding  skeletal remains from the Second World War . Authors confirm that petrous bone outperforms other bone elements in terms of quantity and quality of the recovered DNA and for this reason,  it should always be preferred for genetic testing.

The manuscript is well written but I have a few recommendations that the authors may wish to consider:

-Correlation should be expressed in r, not r2. Considering the above, the discussion should be enriched with research limitations. 

-The resolution of the figures should be higher. 

-I would also like to ask the authors to consider adding a "Conclusions" section which would summarize the most important achievements and limitations of the conducted analyses. Such an additional chapter would significantly enrich this extremely interesting paper. Then, the manuscript would significantly improve in quality. I congratulate the authors on their interesting work. 

Sincerely,

Reviewer

Reviewer 3 Report

Comments and Suggestions for Authors

The primary objective of this study was to establish a DNA extraction method maximizing efficiency for DNA extraction from challenging bone samples and to determine the most suitable bone elements for genetic testing. However, the manuscript predominantly focuses on the overall genetic analysis process of bone remains retrieved from WWII mass graves, within the context of a project, rather than on evaluation of the new experimental method. Consequently, the findings presented may not adequately address the main goal outlined by the authors. It is recommended to reconsider the title to better reflect the manuscript’s content and to provide additional detailed explanations as follows:

Regarding the evaluation of the DNA extraction method:

-       Firstly, to assess the enhanced efficiency of DNA recovery using the ‘new protocol’, it would be effective to compare results obtained from conventional methods, such as the ‘single incubation’ or ‘recommended volume’ methods performed by the authors, rather than solely relying on theoretical comments from the kit manual. (Line 220-228)

-       Additional experimental evidence or clarification is necessary to identify the reasons for the lack of difference in DNA recovery between the results obtained using the Qiagen method in Lab B and those obtained using the proposed method. It remains unclear whether this is attributable to bead capacity limitations or simply methodological differences. (Line229-235)

-       The provided numerical values require review: 1) was ‘380-fold’ calculated from 52.9 ng/0.143 ng or 59.1 ng/0.150 ng? 2) are the median values displayed in figure 3 accurate? (Line 249-253)

-       Concerning the grinding procedure, there is inadequate explanation regarding the differences between the methods employed by Lab A and Lab B for comparison. Additionally, the evidence supporting the “disappointing conclusion that the femurs found in the mass grave actually contained very low amounts of DNA” is insufficient to draw a definitive conclusion based on the results presented in the manuscript. (Line 260-267)

-       Clarification is needed regarding scientific significance of the test for procedure repeatability by different operators three month later. (Line 159-162, Line 268-272) Additionally, determining ‘good repeatability’ should involve considering more potential factors beyond solely relying on the results presented in Figure 1A.

Regarding the appropriate selection of bone elements for DNA extraction:

-       Lines 294-307 appear to belong in the discussion section rather than the results section.

-       Please clarify the y-axis in Figure 4 and the criteria of partial profiles in this study.

Further clarification is required regarding the experimental process:

-       What is the pulverization method utilized by Lab B? How does it differ from the method used by Lab A?

-       Did Lab A employ both the Quantifiler human DNA Quantification kit and the PowerQuant for DNA quantification? (Please clarify this in Figure 1)

-       How were the LOD and ILOQ values calculated in Table 1?

Round 2

Reviewer 1 Report

Comments and Suggestions for Authors

The article is very much improved, and important information was added. However, the authors do not answer my question. So, again, I ask the same:

"Environmental conditions are indeed key to DNA preservation. Authors use this argument to justify the absence of results for teeth samples. However, petrous bones were in the same conditions. How do authors explain that the genetic material from teeth, which are theoretically protected by enamel, did not produce results, and bone samples did?"

The authors' previous answer was: this issue was treated in the conclusions.

In conclusion, their discussion about this is: 

Our results showed that petrous bone -likely due to its intrinsic properties- outperform other bone elements (femur, metacarpals and tooth) in quantity and quality. Overall, the results of this study support that burial conditions play a fundamental role in biomolecule preservation, and no generalisation is allowed: other elements (femur and tooth), which usually provide satisfying results [3,14,33,38,42], are yielding poor results in the case of the mass grave of the isle of Cres. 

This does not answer my question. So I encourage the authors to revise their answer. 

Reviewer 2 Report

Comments and Suggestions for Authors

Dear Authors,

thank you for making the corrections.

Reviewer

Author Response

Thank you for your positive response.